# Barriers to the Adoption of Reverse Logistics in the Construction Industry: A Combined ISM and MICMAC Approach

Margarida Pimentel [1], Amílcar Arantes [2],*[iD] and Carlos Oliveira Cruz [2][iD]

1   Instituto Superior Técnico, Universidade de Lisboa, Av. Rovisco Pais 1, 1049-001 Lisboa, Portugal
2   CERIS, Instituto Superior Técnico, Universidade de Lisboa, Av. Rovisco Pais 1, 1049-001 Lisboa, Portugal
*   Correspondence: amilcar.arantes@tecnico.ulisboa.pt

**Abstract:** With growing environmental concerns, reverse logistics (RL) assumes a significant role in the sustainability of the construction industry to the extent that it can contribute to mitigating some of the negative environmental impacts related to its activity. However, despite the benefits that can be attributed to RL, its implementation level in the construction industry is still very low. This research determines the root barriers to adopting RL in construction (ARLC) using the case of the Portuguese construction market. The methodology involved focus groups and a combined Interpretive Structural Modelling (ISM) and Matrices d'Impacts cross-multiplication appliqúe a classmate (MICMAC) approach. The root barriers that have been identified by the application of the methodology to the ARLC are: lack of financial incentives to incorporate recycled materials, lack of knowledge about RL, lack of technical support, standard codes and regulations in favor of using recycled materials, lack of information sharing, cooperation and coordination among entities of the supply chain, current buildings have not been designed for deconstruction, and lack of construction and demolition waste (CDW) management and recycling infrastructures and markets for the materials resulting from CDW. The highest hierarchical level includes barrier B10 (lack of financial incentives to incorporate recycled materials into the construction); this barrier influences all the other barriers and, as such, it is considered the key barrier to the ARLC in Portugal. The research has also identified 17 different mitigation measures to tackle these barriers, with different natures: fiscal, regulatory, financial, etc.

**Keywords:** reverse logistics; construction and demolition waste; ISM; barriers; mitigation measures

## 1. Introduction

The construction industry contributes significantly to economic growth and social development but has substantial negative environmental effects, particularly in the extraction and consumption of raw materials, water and energy consumption, construction and demolition waste (CDW) production including waste that is harmful to the environment and human health, and soil contamination [1]. Therefore, a large body of knowledge contributes towards increasing the alignment of industry practices with the concepts of sustainability and circular economy [2–7], highlighting the need to reuse CDW.

In the last decade, the construction industry was responsible for producing more than 850 million tons of CDW per year, corresponding to about 36.4% of total waste production [8]. Given this volume, CDW currently constitutes the largest waste stream in the EU. CDW production has been increasing, reaching a maximum of 969 million tons in 2018. CDW is mostly made up of leftovers and the waste of used materials. However, it can be considered that some residues result from small demolition works carried out during the work due to incompatibilities or design errors, and alterations, among others. In this sense, in the reconstruction, alteration, or demolition works, the CDW produced are, for the most part, the materials and constituent elements of the building, removed

or replaced during the execution of the works. In this type of work, the year and time of construction influence the construction techniques and materials used and, consequently, the composition of the CDW produced [9].

Through CDW prevention and reduction, reuse, and recycling, the construction industry can improve its environmental performance, but this requires the establishment of effective and efficient reverse logistics (RL) practices [4,10].

In the construction industry, RL is mainly based on the activities of transport and the storage of CDW produced in construction, reconstruction/alteration, or demolition works, with the objective of reuse and recovery. However, despite the various environmental, economic, and social advantages that can be obtained with the development of RL in construction, there is still a significant lack of knowledge about this concept in the industry, and its level of implementation is still relatively low.

The main objective of this paper is to determine and analyze the root barriers to adopting RL in the Portuguese construction industry and design the respective mitigation measures. The authors applied a combined ISM-MICMAC approach, sustained by experts in focus group discussion, to identify the interrelationships between the root barriers to adopting reverse logistics in construction (ARLC), their hierarchy, and the strengths of those interrelationships [11,12]. As far as the authors are aware, using the combined ISM-MICMAC analysis approach to develop measures to mitigate the barriers to the ARLC represents a novelty in the literature [13,14]. Moreover, the ISM-MICMAC approach was already successfully used to study the adoption of RL in other industries [15–18]. Thus, the present study looks at the Portuguese context and proposes a contribution by identifying and developing mitigation measures for barriers to the ARLC grounded on a combined ISM-MICMAC approach, supported by focus groups, to analyze the complex interrelationships among barriers from a systematic perspective. These measures established to act and impact on the root barriers also impact the other hierarchically dependent barriers, thus effectively impacting the entire system's behavior.

The organization of the paper is the following: firstly, there is a literature review on RL; secondly, the methodology is presented; thirdly, the results are revealed; fourthly, the results are discussed, and measures to mitigate the barriers to the ARLC are proposed; and, finally, conclusions and proposals for future research are presented.

## 2. Background

### 2.1. Definition of Reverse Logistics in the Construction Industry

RL is associated with environmental concerns and efforts to minimize the negative environmental impacts that result from the activities of organizations. From the beginning, its definition refers to recycling and responsible waste disposal. However, there are nuances in the several definitions that can be found in the literature. Stock [19] proposes one of the first definitions for this concept " . . . the term often used to refer to the role of logistics in recycling, waste disposal, and management of hazardous materials; in a broader perspective, it includes all logistical activities related to recycling, substitution, reuse of materials and disposal of waste", as cited in Fleischmann [20].

In 1998, Rogers and Tibben-Lembke [21] defined reverse logistics as "planning, implementing and controlling the efficiency and effectiveness of costs, raw material flows, work in progress and finished products, and related information, from the point of consumption to the point of origin, to recapture value or [perform] proper disposal". For these authors, RL should be more comprehensive than reusing and recycling packaging. According to them, redesigning packaging that requires less material or reducing pollution and energy consumed by transport activities are relevant actions but should be considered in the context of green logistics. Within the scope of reverse logistics, activities such as the remanufacturing and renewal of products should be included. Thus, the authors summarize RL as " . . . the process of moving products from their typical final destination, in the sense of recapturing some value or carrying out its proper disposal" [22]. It should be noted that this definition is still considered valid today and is cited in several recent publications.

Carter and Ellram [23] consider reverse logistics as a way for organizations to improve their environmental performance. These authors define the concept as an inverse distribution of products or materials, in the sense of proceeding with their reuse, recycling, or final disposal, and also include efforts to reduce the supply of products. Thus, by reducing the supply in the forward direction of the distribution channels, the amount of product/material that travels through the reverse channels is also reduced, and both processes become more efficient.

Dowlatshahi [24] refers to RL as a supply chain (SC) that has been redesigned to manage product flows that are destined for remanufacturing, recycling or disposal and allows for more efficient use of resources.

Nunes, Mahler, and Valle [25] define RL as "the area of logistics responsible for planning, managing and controlling the flows of after-sales and post-consumption products, and the respective logistical information in order to add value to them" (economic, environmental, social, among others).

Hosseini et al. [10] explain that the most recent definitions of this concept are more generalized and include activities and procedures that aim to increase the efficiency of the chain and, at the same time, complement the direct logistics system. The authors similarly argue that the purpose of RL is to add value to the entire logistics system. Sellitto [26] mentions that the purpose of RL is to recover part of the value of used products, giving rise to economic, environmental, and social value.

This concept has evolved over the years. Today, RL is seen as fundamental to achieving the sustainable management of materials in line with the principles of the circular economy while opening the way for new opportunities and contributing to the competitiveness of organizations.

Respect for the environment requires organizations to find new ways to prolong materials' useful life and invest in activities that allow them to recover or add some economic, environmental, or social value. In this context, closed-loop or circular SCs have acquired more importance than traditional SCs since they have a great potential for reducing costs for organizations and allow them to minimize costs and environmental impact [27]. Thus, closed-loop SC links forward as does reverse logistics, which includes all processes and activities in the SC forward and backward [28]. Following Guide and Van Wassenhove [27], a closed-loop SC is the process of planning, controlling, and managing a system that makes the most of the value created through the supply cycle and the life of the product with the dynamic recapture of value for different types and volumes over time. These authors argue that this definition does not restrict organizations from focusing on cost efficiency or profitability.

## 2.2. Challenges of RL

The increasing interest that RL has acquired recently can be related to its environmental, economic, and social advantages, among which are diverting the CDW from landfills so they can be reused or recycled, cost savings due to less use of virgin materials, reduced transport and disposal costs, revenue generated by the sale of recovered materials, job creation, and improvement of life quality due to lower pollution levels [29–31]. Additionally, more strict environmental legislation has been implemented, and inevitably, many organizations have been obliged to incorporate RL into their CDW management [29,32]. Closing the loop of materials in the construction industry through the implementation of RL would reduce its environmental impacts while contributing to economic growth. However, it is still impossible to have a completely closed cycle because many materials have a limited number of reuses, and others cannot be reused or recycled and must be disposed of in compliance with the legislation. [2,25].

The SC can be described as a system (or network) of interconnected and interdependent organizations and entities that establish collaborative relationships and information sharing among themselves and whose common interest is to satisfy customer requirements [33].

The entities that make up the SC are interconnected through the material, information, and financial flows [34].

SC management involves coordinating and managing collaborative relationships between the entities/organizations that make up the chain to create more value for the customer with less cost for all parties [33]. SC management must also be based on the interaction between the entities involved, their strategic alignment, the integration of processes and activities, the ability to trust partners, the sharing of information, and the sharing of risks and benefits [35].

*2.3. Barriers to the Adoption of Reverse Logistics*

Hosseini et al. [3] carried out a systematic review of the literature on RL. They identified the main barriers to its implementation, intending to develop a conceptual model highlighting RL's strategic aspects for construction stakeholders. In this sense, the authors classified the barriers as internal and external to the organizations, highlighting the high initial investment and the lack of support within organizations to adopt RL, which can be partially justified by the uncertainty regarding the results and the potential risks, financial and legal costs associated with both the reuse of CDW and the incorporation of recycled materials on-site.

In this context, it should be noted that if, on the one hand, the option of reusing CDW and/or incorporating recycled materials is a way of reducing the environmental impacts of the industry, on the other hand, they require more time and flexibility on the part of the responsible designers and contractors throughout the project, and may still represent a higher risk, since it may not be possible to guarantee the quality and/or performance of these materials, and materials that meet the project specifications may not be available on the market [36].

To incorporate RL into the management of CDW, the decision to deconstruct a building is considered as a starting point, since it is a phased and controlled demolition method, which makes it possible to recover materials and constructive elements, while minimizing the risk of damage and contamination by hazardous substances, and optimizes their reuse or recycling. However, compared to traditional demolition, deconstruction is a more time-consuming process and, due to the nature of the work involved, it implies higher labor costs and more working space.

Regarding external barriers, the authors distinguish two categories: barriers related to the singularities of the construction industry and barriers related to the specific characteristics of the final products of construction (e.g., buildings).

In this sense, barriers related to the industry include the lack of infrastructure for the management and recycling of CDW, the industry's lack of awareness concerning RL, the lack of technical support, regulations, and financial incentives for the incorporation of recycled materials in construction, the adverse social perception concerning the quality of CDW and recycled materials, and the low rates of CDW deposition in landfills not rewarding the costs associated with RL.

In the barriers related to the final products of the construction, mention is made of their size and immobility, the long lifespan of buildings, with multiple owners, and maintenance/renovation works, which make it challenging to know the materials and processes used in the building to be deconstructed, and the variability in terms of quality, size, level of deterioration, and hazardousness of recovered CDW.

It should be noted that the size and immobility of these products mean that, during the demolition phase, the materials and elements that make up the building have to be extracted and collected on-site and then transported to their respective destinations. This translates into multiple and uncertain CDW origin points, which correspond to the location of the works, and a few destination points, which correspond to CDW management operators. These factors constitute challenges in planning and managing RL activities, specifically in planning and optimizing routes, which depend on the location of the works and their

respective access conditions. In many cases, the only viable solution for transporting CDW to their final destinations is truck transport, subject to weight and size restrictions [37].

Hosseini et al. [3] also mention one more relevant aspect, that most of the buildings that are currently the target of deconstruction were not designed/conceived for this purpose. This means that the selection of construction methods and materials used, among others, did not consider their future dismantling, which inevitably makes this process more time-consuming, expensive, and complex.

Chileshe et al. [2] used a mixed methods approach to determine and rank the critical barriers to implementing RL in the construction industry, focusing on building construction and demolition projects. These authors analyzed 16 barriers, which they characterized in terms of operational aspects, aspects of the construction industry, and social aspects. Regarding the operational aspects, it is highlighted that the deconstruction of a building usually takes longer and includes activities considered to be of high risk in terms of the health and safety of workers, as well as requiring experienced and specialized labor, which translates into higher costs compared to traditional demolition. However, the fact that the buildings were not designed for deconstruction exacerbates the impact of some of these aspects.

In this context, it should be noted that although there are some similarities between traditional demolition and deconstruction, in traditional demolition operations there is no concern to separate the CDW at source according to their characteristics, which results in the production of large quantities and volumes of mixed materials that end up in landfills. In deconstruction, the purpose is the careful and controlled dismantling of the building to maximize the number and quantity of materials and constituent elements recovered in conditions to be reused or recycled. The separation of these materials is carried out at the source to minimize contamination by hazardous substances. Thus, as expected, in deconstruction, the level of planning is higher since it is necessary to define, among others, the demolition methods and techniques to be used, the actors and their respective responsibilities, and also to identify all the waste that will be produced during the works, as well as their respective quantities and appropriate final destinations [2]. These authors also mention the lack of guarantee regarding the availability on the market of CDW and/or recycled materials that meet the project's requirements as an operational difficulty.

In aspects related to the construction industry, Chileshe et al. [2] highlight some relevant constraints, namely, their non-incorporation into the work by designers, the lack of instruments that allow the validating of the quality of CDW/recycled materials, the higher prices of materials from CDW compared to new materials, the perception of negative social impact regarding their use, and they also add that current regulations restrict or make their use complicated.

On the social aspects, Chileshe et al. [2] mention that local regulatory authorities and municipal inspectors do not promote/support deconstruction and that there is also disapproval from the work inspectors and end-users regarding the incorporation of CDW/recycled materials. The authors also add the condition of legal liability that companies entail related to their use of CDW and recycled materials.

In this study, the authors concluded that the barriers with the most significant influence on the implementation of RL are: the non-incorporation of CDW and recycled materials on-site by the designers; current regulations restricting the use of these materials; the legal liability that companies incur when they choose to use these materials; and also the cost and duration of deconstruction compared to traditional demolition.

Rameezdeen et al. [38] conducted a qualitative study supported by interviews to identify the construction industry's critical barriers in implementing RL practices and their respective interrelationships. In this way, the authors analyzed 12 barriers, namely, the current norms and regulations, the additional costs associated with RL, the lack of knowledge about RL in the industry, the need for more workforce, the associated financial risks, contractual obligations, the unavailability of designers to conceive/design for deconstruction, the lack of support from end-users in the incorporation of CDW and re-

cycled materials on-site, resistance to change on the part of construction stakeholders, the health of and increased security for workers, the lack of support information systems and CDW management infrastructures, and the lack of information sharing between the chain's actors.

In this sense, concerning current norms and regulations, the authors highlight their lack of incorporation of CDW/recycled materials on-site and the low rates for deposition of CDW in landfills, which does not encourage the search for alternative solutions, in particular, reuse and recycling. The additional costs relate mainly to the labor costs necessary to carry out the activities inherent to the deconstruction, the transport of CDW, and the higher price of recycled materials, compared to new materials, which does not promote their use.

The lack of knowledge about RL in the industry is the responsibility of all actors in the construction processes. However, despite the degree of responsibility not being identical, everyone should be aware of the importance of implementing RL. As an example, Rameezdeen et al. [38] mention that most designers do not consider RL when designing their projects (design for deconstruction, reuse of CDW, incorporation of recycled materials, among others) and that many contractors not only disapprove of the option of using CDW/recycled materials, as well, they are not open to changing their current practices to accommodate RL. In fact, there is much resistance to changing behavior and adopting innovations in the construction industry, and it is not exclusive to RL. However, concerning recycled materials, the lack of support/acceptance from end-users also does not encourage their incorporation into the work.

The increased effort to implement RL is related to the planning and execution of activities, such as source separation and CDW transport, and the performance of specific procedures when there are hazardous materials. Due to the nature of the activities involved, there is also expected to be an increased health and safety risk for workers compared to traditional demolition.

In this context, it is also important to highlight that reusing CDW and incorporating recycled materials in work requires an increased planning effort in the project design phase and a greater willingness to look for materials that correspond to the defined specifications. It also requires greater flexibility on the part of designers and contractors, given the need for unexpected changes, to accommodate the unavailability of materials and errors in quality, size, and quantity. By comparison, this extra effort is unnecessary when virgin materials are used in the project [36].

Current CDW management practices have known risks and benefits. At the same time, the advantages related to the implementation of RL are not yet perceptible to the various stakeholders, which can generate a notion of risk associated with ignorance. However, the fact that there is no guarantee of the quality and safety of the materials resulting from CDW prevents financial and legal risks from being determined in advance, not encouraging their use on-site. Also, contractual obligations, in the usual terms in which they are celebrated, with very short deadlines for the demolition phase, encourage traditional practices to the detriment of deconstruction.

Finally, the authors mention the lack of infrastructure for the management and recycling of CDW, the lack of information systems to support RL, and the lack of information sharing between the partners in the chain. These last two constraints were later analyzed in other works by the same authors [39,40].

Rameezdeen et al. [38] conclude that standards and regulations, additional costs, lack of knowledge about RL, and the increased effort for its implementation are critical barriers to the ARLC. However, they add that the lack of knowledge, rules, and regulations, and workers' health and safety risks are the barriers that are more interrelated with others, so they are the best candidates on which to apply mitigation measures.

Ambekar et al. [13] used the Delphi Method to determine the main barriers to implementing RL in construction. Then they used an ISM approach to study the interrelationships between these barriers and a MICMAC analysis to classify them according to their dependence and driving powers. These authors identified and analyzed ten barriers, which are:

lack of knowledge about RL in construction, namely, about the associated economic, environmental, and social benefits; the lack of perception, specifically concerning the economic advantages for organizations; the lack of government policies, considered fundamental to encourage and promote the adoption of RL; the lack of appropriate information systems to support RL; the lack of standards/technical support to incorporate CDW/recycled materials on-site; the lack of support from the industry's stakeholders; lack of resources and financial capacity; inadequate corporate policies to adopt RL; the lack of training in the recycling and reuse of CDW; and very rigid decision-making processes within the industry's organizations, which do not facilitate the implementation of RL.

## 3. Methods and Materials

Human behavior or actions are present in most construction project phases, suggesting that effective construction research requires appropriate social science research methods. These methods suit studies that include human actions or behavior in construction projects, such as innovation, planning, and leadership [41]. Therefore, ISM, MICMAC analysis, and focus groups (FG) are considered valuable for researching the barriers to the ARLC. The methodological approach adopted in this research comprises four phases, as presented in Figure 1: Phase I—Determination of the critical barriers, Phase II—ISM, Phase III—MICMAC analysis, and Phase IV—Development of mitigation measures.

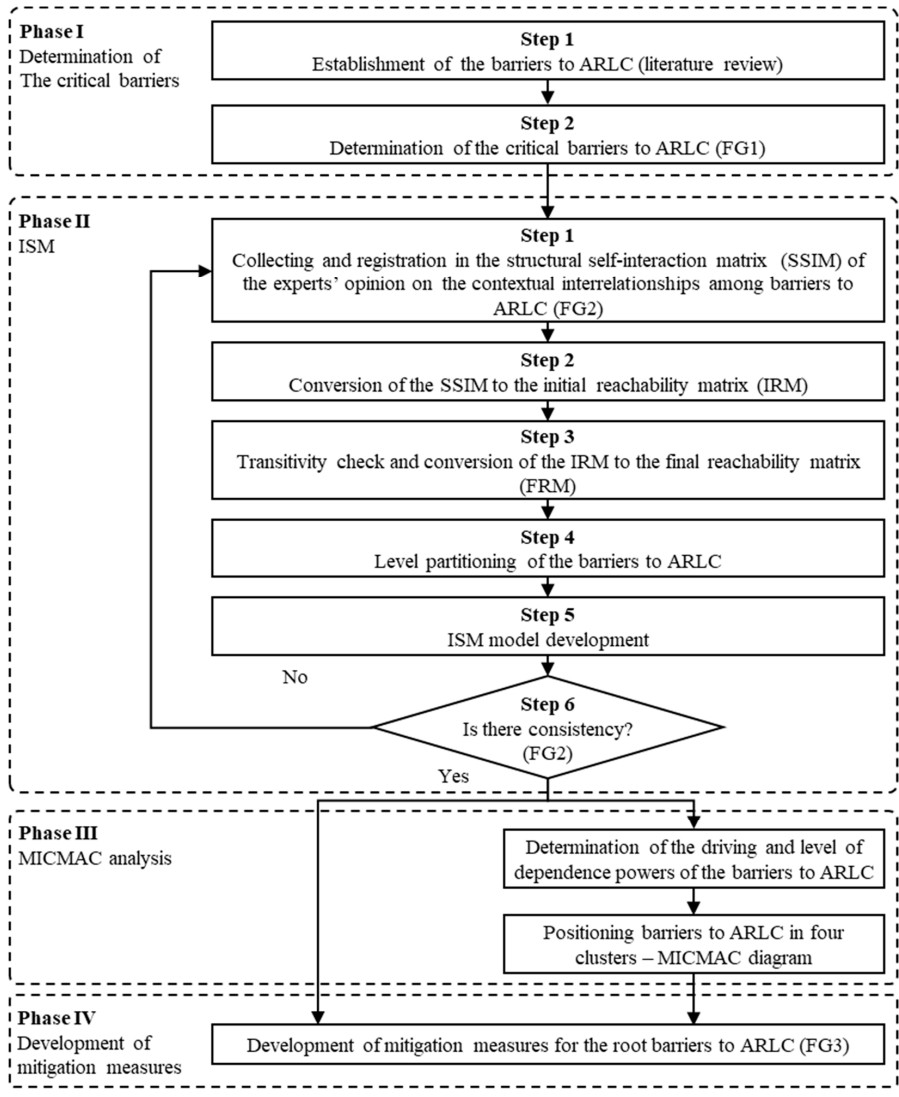

**Figure 1.** Research methodology.

The FG is a research method of exploratory nature that collects qualitative data from group interaction on a topic presented by the moderator [42]. The FG stimulates discussion among experts concerning their insights, opinions, attitudes, and beliefs regarding a product, service, theory, or concept, enlarging and improving the information existing on a topic and creating new insights [43].

Nassar-McMillan and Borders [44] recommend that FGs comprise four to twelve participants [45]. In this study, nine experts were used for the three FGs. Six of them were practitioners with ten or more years of experience in construction and some knowledge of RL; three were academics, two were in Civil Engineering, and one was in the area of SC management in construction, all with an average experience of 15 years. To minimize the bias effect, practitioners included clients, consultants, and contractors in equal numbers.

The three FGs occurred between October and November 2021, with a two-week interval. The first FG (FG1) selected the critical barriers, the second (FG2) defined the interrelationships among barriers and checked the ISM model consistency, and the last one (FG3) assisted in defining the measures to mitigate the barriers to the ARLC. FG1, FG2, and FG3 lasted about one and a half, two, and one hours, respectively. FG1 and FG2 were in person, and FG3 was via Microsoft Teams. The experts in FG1 were educated on the RL in construction and given a brief presentation of the barriers to the ARLC and the rationale of the ISM-MICMAC approach.

The FGs were moderated by one of the authors, safeguarding a clear understanding of RL and the barriers to the ARLC. He assisted in reaching a broad consensus, encouraged debate, and guaranteed the argument advanced from wide to specific issues to boost honesty and lessen bias [46]. Moreover, he used the principle of "majority rules" [47] whenever, occasionally, experts had differences in opinions.

### 3.1. Phase I: Identification of the Critical Barriers to the ARLC

Phase I encompasses two steps described in the methodology (Figure 1).

Step 1: Establishment of the barriers to the ARLC

Based on the studies mentioned in the literature review section, in particular Chileshe et al. [2], Hosseini et al. [3], Rameezdeen et al. [38], Tennakoon et al. [39], Wijewickrama et al. [40], Ambekar et al. [13], and Govindan et al. [48], the researchers selected and adapted the barriers to the ARLC for the Portuguese context, making it possible to tackle worries about some issues being "lost in translation" [49]. In addition, special care was taken to avoid vague words and doubts to guarantee reliable meaning for experts and augment their opinions' validity. The resulting 32 barriers to the ARLC are presented in Table 1.

**Table 1.** List of barriers to the ARLC.

| No. | Barriers |
| --- | --- |
| 1 | High costs associated with RL |
| 2 | The lack of support or support within the organization |
| 3 | Business policies/models are inadequate |
| 4 | Unavailability by designers/architects to use RCW or incorporate recycled materials |
| 5 | Unavailability on the part of the designers in designing for deconstruction |
| 6 | Resistance to change or innovation from industry stakeholders |
| 7 | Decision processes within organizations are very strict |
| 8 | Lack of knowledge about RL's potential economic, environmental and social benefits |
| 9 | Lack of training in recycling and reuse of CDW |
| 10 | Lack or low use of support information systems |
| 11 | Lack of knowledge about RL in the construction industry |
| 12 | Lack of information sharing, cooperation, and coordination among SC entities |
| 13 | Lack of CDW management and recycling infrastructure and markets for the materials resulting from CDW |
| 14 | Lack of approval by the consultants for the use of CDW or recycled materials |
| 15 | Existing buildings were not designed for deconstruction |
| 16 | The long lifetime of buildings—with multiple owners throughout their life and maintenance/renovation works—makes it challenging to know the materials and processes used in the building to be rebuilt |

**Table 1.** *Cont.*

| No. | Barriers |
|---|---|
| 17 | Contractual obligations do not provide the possibility of deconstruction/use of CDW/recycled materials |
| 18 | Compared with traditional demolition, deconstruction requires more construction space |
| 19 | Compared with traditional demolition, deconstruction contributes to delays |
| 20 | Compared with traditional demolition, deconstruction presents greater safety and health risks for workers |
| 21 | Compared to traditional demolition, deconstruction requires superior control and management effort |
| 22 | Deconstruction requires more experienced and qualified personnel compared to traditional demolition |
| 23 | Large variety in quality, dimension, level of deterioration, and level of hazard of CDW |
| 24 | The immobility of construction products and multiple and dispersed CDW points of origin translates into inefficient transport routes. |
| 25 | Availability (lack of) CDW/recycled materials in markets |
| 26 | Lack of technical support, standard codes, and regulations in favor of using recycled materials |
| 27 | Lack of financial incentives to incorporate recycled materials into the construction |
| 28 | CDW landfill rates are meager (which does not justify the costs associated with RL) |
| 29 | Lack of warranty or certification regarding the quality of CDW/recycled materials |
| 30 | Local regulatory authorities/municipal tax authorities do not promote/support deconstruction |
| 31 | Current regulations restrict and hinder the use of CDW and recycled materials |
| 32 | Negative social perception of the quality of CDW/recycled materials |

Step 2: Determination of the critical barriers to the ARLC

Firstly, the moderator in FG1 presented and discussed the 32 barriers to the ARLC presented in Table 1 with the experts. To guide the experts, the moderator started by asking them if the barriers were appropriate, if there were resemblances between them regarding the nature of this research, and if any other barriers were absent. Then, the moderator demanded that the experts select the barriers they judged critical to the ARLC and that should be considered in this research. After gathering the experts' responses, the research team analyzed them. A barrier was selected only when the experts were unanimous, or the majority agreed on its criticality in the Portuguese context. The experts selected 12 critical barriers (Table 1). This number is similar to those used in comparable studies [50–52] and below the recommendation from Wu et al. [53] of 15, limiting the time experts in the FG2 spend assessing the interrelationships among barriers. The critical barriers to the ARLC in Portugal, from now on designated simply by barriers, are presented in Table 2.

**Table 2.** List of critical barriers to the ARLC in Portugal.

| Code | Critical Barriers | Description |
|---|---|---|
| B1 | Compared to traditional demolition, deconstruction requires superior control and management effort | There is a lack of expertise related to deconstruction. Deconstruction is a process characterized by systematically dismantling a structure and its elements, through controlled demolition techniques, in the opposite direction to its construction process [4]. |
| B2 | Lack of CDW management and recycling infrastructures and markets for the materials resulting from CDW | Suppose there are no management and recycling infrastructure of CDW and markets within a certain distance of the construction site. In that case, transport costs become too high, and the possible recovery and reuse of CDW may no longer be economically viable [3,38,54]. |
| B3 | Existing buildings were not designed for deconstruction | The buildings currently being deconstructed were not designed for this purpose, which means that the construction techniques and materials used, among other aspects, did not consider their easy deconstruction. Thus, their deconstruction becomes more time-consuming, expensive, and complex [2,3]. |
| B4 | High costs associated with RL | The implementation of RL in construction has high costs, mainly associated with the labor required to deconstruct buildings, sort, store, and transport CDW [2,38,55]. |
| B5 | Lack of warranty or certification regarding the quality of CDW/recycled materials | The lack of quality warranty or certification of materials from CDW results in a lack of confidence regarding their properties and performance, which inevitably restricts their demand and incorporation in work [2]. |

**Table 2.** *Cont.*

| Code | Critical Barriers | Description |
|------|-------------------|-------------|
| B6 | Resistance to change or innovation from industry stakeholders | In the construction industry, there is resistance to implementing changes or innovations [38], which makes adopting and implementing RL difficult [29,56]. |
| B7 | Lack of information sharing, cooperation, and coordination among SC entities | The characteristics of the construction industry, in particular the fragmented relationships between the various entities involved in the construction SC, contribute to poor information sharing, cooperation, and coordination, which results in a decrease in the efficiency and effectiveness of the RL [39,57,58]. |
| B8 | Deconstruction requires more experienced and qualified personnel compared to traditional demolition | Experienced and qualified personnel are essential to improve the quantity and quality of the CDWs obtained, enhancing their future valorization and minimizing their exposure to risks [2,38,59]. |
| B9 | Lack of technical support, standard codes, and regulations in favor of using recycled materials | The existing technical standards and regulations regarding incorporating recycled materials in construction are currently insufficient and restrictive, with only a few uses for these materials in the industry [3,38]. |
| B10 | Lack of financial incentives to incorporate recycled materials into the construction | To promote changes in behavior and the adoption of innovations in the construction industry (in the present case, CDW-related materials), it is essential that governments develop incentives, both in regulatory and economic-financial terms, to stimulate the demand for and the incorporation of recycled materials in the construction site [3,60]. |
| B11 | Variety in quality, dimension, level of deterioration, and hazard of CDW | The CDW's composition (variability and quality) depends on several factors, including construction techniques, the materials used, the age of the building, and the alteration or renovation works carried out during its useful life [3,4,61]. |
| B12 | Lack of knowledge about RL in the construction industry | Although the literature on RL in the construction industry has recently increased, the industry's knowledge of RL is still low [14,16]. |

### 3.2. Phase II: ISM

Warfield first presented the ISM method in 1974 [62]. ISM is a learning group process aided by a computer and used to research the interrelationships among variables (barriers in the present work) regarding a subject of a specific multifaceted system [63]. It deciphers unclear conceptual models into visible and unequivocal systems, increasing the understanding of their variables by defining their interrelationships and hierarchy [64]. Moreover, it designates the variables at the top of the hierarchy as the key variables that control the system's behavior.

Through the knowledge and experience of experts, the ISM puts forward the awareness of the interrelationships between variables and helps researchers understand how the hierarchy is established between these variables, thus defining the level and direction of complex interrelationships between system variables [11,12]. Furthermore, ISM integrates experts' opinions and permits them to review their opinions and modify evaluations, providing the opportunity to be used in real-life systems [65].

Step 1: Contextual interrelationships among barriers

The 12 critical barriers to the ARLC generated 105 (12 × 11/2 = 66) different interrelationships. The experts in FG2 were requested to express the contextual interrelationships among a pair of barriers (Bi and Bj), giving four different types:

- V: Bi helps to achieve or influences Bj;
- A: Bj helps to achieve or influences Bi;
- X: Bi helps to achieve or influences Bj and vice versa;
- O: no interrelationship exists among Bi and Bj.

The resulting Structural Self-Intersection Matrix (SSIM) is presented in Table 3.

**Table 3.** Structural self-intersection matrix.

| Bi [1] ↓, Bj [2] → | B1 | B2 | B3 | B4 | B5 | B6 | B7 | B8 | B9 | B10 | B11 | B12 |
|---|---|---|---|---|---|---|---|---|---|---|---|---|
| B1 | | O | A | O | X | O | A | V | O | O | A | O |
| B2 | | | O | V | O | O | O | O | O | A | O | A |
| B3 | | | | V | O | V | V | V | O | A | V | A |
| B4 | | | | | V | O | A | A | A | A | A | A |
| B5 | | | | | | A | A | O | A | A | A | A |
| B6 | | | | | | | O | O | A | O | A | O |
| B7 | | | | | | | | O | A | O | O | X |
| B8 | | | | | | | | | O | O | X | O |
| B9 | | | | | | | | | | O | V | A |
| B10 | | | | | | | | | | | O | V |
| B11 | | | | | | | | | | | | O |
| B12 | | | | | | | | | | | | |

[1] Bi—barrier in line i; [2] Bj—barrier in column j.

Step 2: Conversion of the SSIM into the initial reachability matrix

The SSIM was converted into the initial reachability matrix (IRM), a matrix that represents the direct interrelationships among barriers, replacing V, A, X, and O with 1s and 0s given to the rules in Table 4.

**Table 4.** Rules to convert SSM into IRM.

| SSIM (i, j) | IRM | |
|---|---|---|
| | (i, j) | (j, i) |
| V | 1 | 0 |
| A | 0 | 1 |
| X | 1 | 1 |
| O | 0 | 0 |

Step 3: Conversion of the IRM into the final reachability matrix

The transitivity of the IRM was checked, resulting in the Final Reachability Matrix (FRM). If barrier i influences barrier j and if barrier j influences barrier k, then barrier i indirectly influences barrier k, and if IRM (i,k) = 0, then FRM (i,k) = 1*. Before computing the FRM, the identity matrix was added to the IRM, resulting in the matrix *IIRM*. The FRM was then computed according to the following Boolean operation [53]:

$$IIRM \neq IIRM^2 \ldots \neq IIRM^{n-1} \neq IIRM^n = IIRM^{n+1} = RFM \qquad (1)$$

Step 4: Level partitioning

The level partitioning of the FRM is an iterative process carried out to define the hierarchy among barriers. First, the reachability, antecedent, and intersection sets (RS, AS, and IS) were generated for each barrier to assess its partition level (hierarchical level). The RS of barrier i comprises the barriers that are influenced by barrier i (i.e., barriers j indicated by 1s along row i). The AS of barrier j comprises the barriers that influence barrier j (i.e., barriers i indicated by 1s along column j). The IS comprises the barriers resulting from the RS and AS interception. If an IS and RS of a barrier were identical, then the barrier was attributed to the iteration level (partition/hierarchical level). All the barriers attributed to a given iteration level were detached from the RS and AS, and the process advanced to the following iteration. The process ended when all barriers were assigned to their partition/hierarchical level.

Step 5: ISM model development

The ISM model is a digraph made from the FRM and the level partitioning of the barriers. First, to help structure the ISM model, the FRM conic matrix was created, grouping barriers by hierarchical level. Then, an initial digraph was prepared, positioning the barriers

vertically by their partition levels, connecting them through arrows, and considering the conical matrix. Finally, the indirect connections among barriers at different levels were deleted to obtain the ISM model.

Step 6: Consistency verification

Finally, the ISM model of the barriers to the ARLC was verified by the experts at FG2 for conceptual consistency, and corrections were made if necessary.

### 3.3. Phase III: MICMAC Analysis

Duperrin and Godet [66] established the MICMAC analysis founded on the multiplication properties of matrices. MICMAC analysis helps classify and understand the barriers to the ARLC, giving their driving power (DVP) and dependence power (DPP) [65]. The DVP is the barrier's ability to affect the other barriers, and the DPP is the extent to which other barriers affect it [47]. The DVP of barrier i is the sum of 1 s on line i, and the DPP is the sum of 1 s on column i of the FRM. Next, a DVP-DPP diagram was created, and the barriers were positioned in one of the clusters given to their DVP and DPP, namely: autonomous (low DVP and low DPP), independent (high DVP and low DPP), linkage (high DVP and high DPP), and dependent (low DVP and high DPP).

### 3.4. Phase IV: Development of Mitigation Measures

In phase IV of the research methodology, measures to mitigate barriers to the ARLC were established with the help of FG3 experts. Initially, the experts interpreted the results from the ISM model and the MICMAC diagram. Then, they were invited to propose measures to mitigate the impacts of the key barriers. Finally, they were instructed to verify whether the measures proposed effectively mitigate all the other barriers; otherwise, they should recommend complementary measures. This procedure should lead to the development of practical and effective measures to mitigate the barriers to the ARLC, once they are established considering the hierarchy observed of the interrelationships among the barriers and their DVP and DPP and are grounded on the experience of the experts.

## 4. Results

This section presents the development of the ISM model and the MICMAC analysis.

### 4.1. ISM Model

The ISM model was developed following steps 1 to 6 of phase II of the research methodology (Figure 1). In step 2, the SSIM was transformed into the IRM (Table 5) using the rules in Table 4. If the IRM(i,j) is 1 this means that barrier i helps to achieve or influences barrier j; if it is 0, there is no interrelationship among the barriers i and j.

**Table 5.** Initial reachability matrix (IRM).

| Bi [1] ↓, Bj [2] → | B1 | B2 | B3 | B4 | B5 | B6 | B7 | B8 | B9 | B10 | B11 | B12 |
|---|---|---|---|---|---|---|---|---|---|---|---|---|
| B1 | 1 | 0 | 0 | 0 | 1 | 0 | 0 | 1 | 0 | 0 | 0 | 0 |
| B2 | 0 | 1 | 0 | 1 | 0 | 0 | 0 | 0 | 0 | 0 | 0 | 0 |
| B3 | 1 | 0 | 1 | 1 | 0 | 1 | 1 | 1 | 0 | 0 | 1 | 0 |
| B4 | 0 | 0 | 0 | 1 | 1 | 0 | 0 | 0 | 0 | 0 | 0 | 0 |
| B5 | 1 | 0 | 0 | 0 | 1 | 0 | 0 | 0 | 0 | 0 | 0 | 0 |
| B6 | 0 | 0 | 0 | 0 | 1 | 1 | 0 | 0 | 0 | 0 | 0 | 0 |
| B7 | 1 | 0 | 0 | 1 | 1 | 0 | 1 | 0 | 0 | 0 | 0 | 1 |
| B8 | 0 | 0 | 0 | 1 | 0 | 0 | 0 | 1 | 0 | 0 | 1 | 0 |
| B9 | 0 | 0 | 0 | 1 | 1 | 1 | 1 | 0 | 1 | 0 | 1 | 0 |
| B10 | 0 | 1 | 1 | 1 | 1 | 0 | 0 | 0 | 0 | 1 | 0 | 1 |
| B11 | 1 | 0 | 0 | 1 | 1 | 1 | 0 | 1 | 0 | 0 | 1 | 0 |
| B12 | 0 | 1 | 1 | 1 | 1 | 0 | 1 | 0 | 1 | 0 | 0 | 1 |

[1] Bi—barrier in line i; [2] Bj—barrier in column j.

In step 3, the IRM was transformed into the FRM (Table 6) through the transitivity check with a Microsoft VBA routine. The DVP and DPP of each barrier were also calculated and presented in the last column (sum along the respective line) and line (sum along the respective column) of the FRM, respectively. An entry (i,j) in the FRM equal to 1* indicates that the interrelationship between Bi and Bj results from the transitivity check.

**Table 6.** Final reachability matrix (FRM).

| Bi [1] ↓, Bj [2] → | B1 | B2 | B3 | B4 | B5 | B6 | B7 | B8 | B9 | B10 | B11 | B12 | DVP [4] |
|---|---|---|---|---|---|---|---|---|---|---|---|---|---|
| B1 | 1 | 0 | 0 | 1* | 1 | 1* | 0 | 1 | 0 | 0 | 1* | 0 | 6 |
| B2 | 1* | 1 | 0 | 1 | 1* | 1* | 0 | 1* | 0 | 0 | 1* | 0 | 7 |
| B3 | 1 | 1* | 1 | 1 | 1* | 1 | 1 | 1 | 1* | 0 | 1 | 1* | 11 |
| B4 | 1* | 0 | 0 | 1 | 1 | 1* | 0 | 1* | 0 | 0 | 1* | 0 | 6 |
| B5 | 1 | 0 | 0 | 1* | 1 | 1* | 0 | 1* | 0 | 0 | 1* | 0 | 6 |
| B6 | 1* | 0 | 0 | 1* | 1 | 1 | 0 | 1* | 0 | 0 | 1* | 0 | 6 |
| B7 | 1 | 1* | 1* | 1 | 1 | 1* | 1 | 1* | 1* | 0 | 1* | 1 | 11 |
| B8 | 1* | 0 | 0 | 1 | 1* | 1* | 0 | 1 | 0 | 0 | 1 | 0 | 6 |
| B9 | 1* | 1* | 1* | 1 | 1 | 1 | 1 | 1* | 1 | 0 | 1 | 1* | 11 |
| B10 | 1* | 1 | 1 | 1 | 1 | 1* | 1* | 1* | 1* | 1 | 1* | 1 | 12 |
| B11 | 1 | 0 | 0 | 1 | 1 | 1 | 0 | 1 | 0 | 0 | 1 | 0 | 6 |
| B12 | 1* | 1 | 1 | 1 | 1 | 1* | 1 | 1* | 1 | 0 | 1* | 1 | 11 |
| DPP [3] | 12 | 6 | 5 | 12 | 12 | 12 | 5 | 12 | 5 | 1 | 12 | 5 | |

[1] Bi—barrier in line i; [2] Bj—barrier in column j; [3] DPP—Dependence power; [4] DVP—Driving power; 1*—Transitive interrelationships.

In step 4, the FRM partitioning process took four iterations; that is, the barriers to the ARLC are distributed in four hierarchical levels (Table 7).

**Table 7.** Level partitioning of the FRM.

| Barrier | Reachability Set | Antecedent Set | Intersection Set | Level |
|---|---|---|---|---|
| B1 | B: 1, 4, 5, 6, 8, 11 | B: 1, 2, 3, 4, 5, 6, 7, 8, 9, 10, 11, 12 | B: 1, 4, 5, 6, 8, 11 | I |
| B4 | B: 1, 4, 5, 6, 8, 11 | B: 1, 2, 3, 4, 5, 6, 7, 8, 9, 10, 11, 12 | B: 1, 4, 5, 6, 8, 11 | I |
| B5 | B: 1, 4, 5, 6, 8, 11 | B: 1, 2, 3, 4, 5, 6, 7, 8, 9, 10, 11, 12 | B: 1, 4, 5, 6, 8, 11 | I |
| B6 | B: 1, 4, 5, 6, 8, 11 | B: 1, 2, 3, 4, 5, 6, 7, 8, 9, 10, 11, 12 | B: 1, 4, 5, 6, 8, 11 | I |
| B8 | B: 1, 4, 5, 6, 8, 11 | B: 1, 2, 3, 4, 5, 6, 7, 8, 9, 10, 11, 12 | B: 1, 4, 5, 6, 8, 11 | I |
| B11 | B: 1, 4, 5, 6, 8, 11 | B: 1, 2, 3, 4, 5, 6, 7, 8, 9, 10, 11, 12 | B: 1, 4, 5, 6, 8, 11 | I |
| B2 | B2 | B: 2, 3, 7, 9, 10, 12 | B2 | II |
| B3 | B: 3, 7, 9, 12 | B: 3, 7, 9, 10, 12 | B: 3, 7, 9, 12 | III |
| B7 | B: 3, 7, 9, 12 | B: 3, 7, 9, 10, 12 | B: 3, 7, 9, 12 | III |
| B9 | B: 3, 7, 9, 12 | B: 3, 7, 9, 10, 12 | B: 3, 7, 9, 12 | III |
| B12 | B: 3, 7, 9, 12 | B: 3, 7, 9, 10, 12 | B: 3, 7, 9, 12 | III |
| B10 | B10 | B10 | B10 | IV |

In step 5, the ISM model presented in Figure 2 was developed with the assistance of the FRM conical matrix (Table 8). Lastly, in step 6, the experts in FG2 were requested to verify the consistency of the model. The experts concluded that the ISM model was consistent with their mental model of the system of barriers to the ARLC. Therefore, the ISM model was considered adequate, emphasizing the hierarchy and the interrelationships of the barriers to the ARLC.

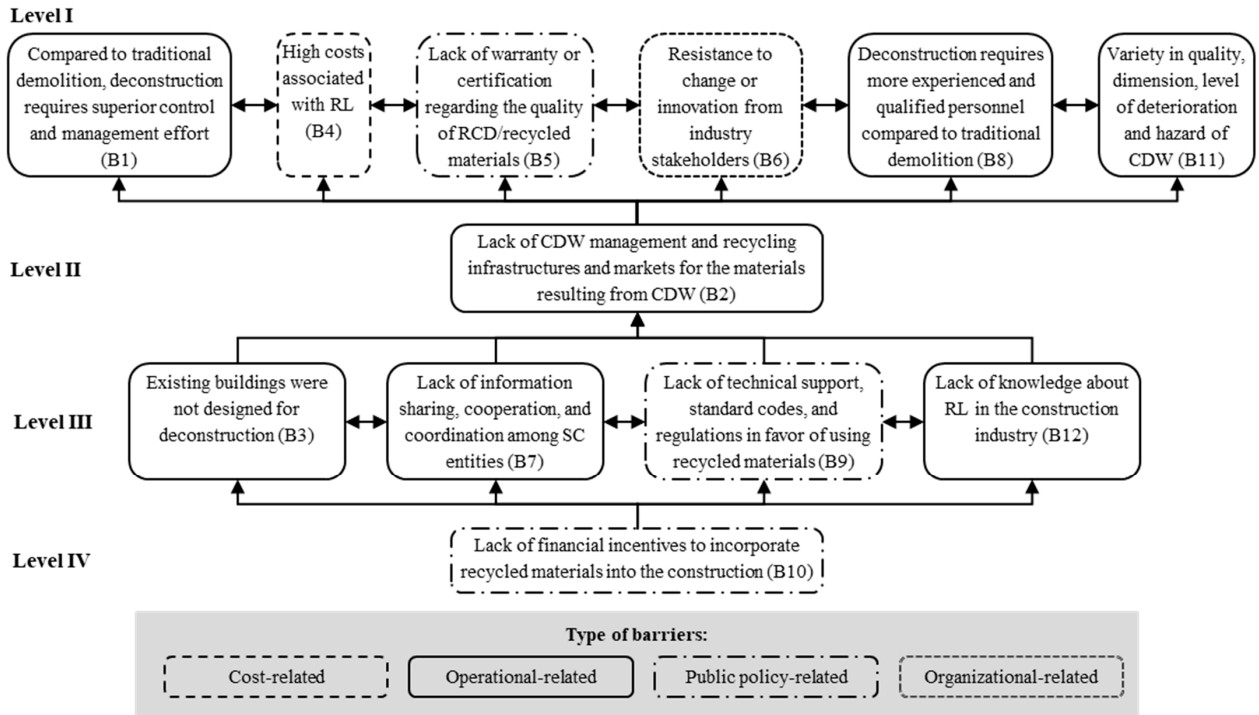

**Figure 2.** The ISM model of the barriers to the ARLC in Portugal.

**Table 8.** The conical Final Reachability Matrix.

| Bi [1] ↓, Bj [2] → | B1 | B4 | B5 | B6 | B8 | B11 | B2 | B3 | B7 | B9 | B12 | B10 |
|---|---|---|---|---|---|---|---|---|---|---|---|---|
| B1 | I | 1 | 1 | 1 | 1 | 1 | 0 | 0 | 0 | 0 | 0 | 0 |
| B4 | 1 | I | 1 | 1 | 1 | 1 | 0 | 0 | 0 | 0 | 0 | 0 |
| B5 | 1 | 1 | I | 1 | 1 | 1 | 0 | 0 | 0 | 0 | 0 | 0 |
| B6 | 1 | 1 | 1 | I | 1 | 1 | 0 | 0 | 0 | 0 | 0 | 0 |
| B8 | 1 | 1 | 1 | 1 | I | 1 | 0 | 0 | 0 | 0 | 0 | 0 |
| B11 | 1 | 1 | 1 | 1 | 1 | I | 0 | 0 | 0 | 0 | 0 | 0 |
| B2 | 1 | 1 | 1 | 1 | 1 | 1 | II | 0 | 0 | 0 | 0 | 0 |
| B3 | 1 | 1 | 1 | 1 | 1 | 1 | 1 | III | 1 | 1 | 1 | 0 |
| B7 | 1 | 1 | 1 | 1 | 1 | 1 | 1 | 1 | III | 1 | 1 | 0 |
| B9 | 1 | 1 | 1 | 1 | 1 | 1 | 1 | 1 | 1 | III | 1 | 0 |
| B12 | 1 | 1 | 1 | 1 | 1 | 1 | 1 | 1 | 1 | 1 | III | 0 |
| B10 | 1 | 1 | 1 | 1 | 1 | 1 | 1 | 1 | 1 | 1 | 1 | IV |

[1] Bi—barrier in line i; [2] Bj—barrier in column j. The hierarchical levels are placed diagonally across the matrix, and the shadow demarcates the barriers at each level.

Level I, at the top of the ISM model and the lowest hierarchical level, comprises the barriers B1 (deconstruction requires superior control and management effort), B4 (high costs associated with RL), B5 (lack of warranty or certification regarding the quality of CDW/recycled mate), B6 (resistance to change or innovation from industry stakeholders), B8 (deconstruction requires more experienced and qualified personnel) and B11 (variety in quality, dimension, level of deterioration and hazard of CDW). Level II comprises only barrier B2 (lack of CDW management and recycling infrastructures and markets for the materials resulting from CDW). Level III includes four barriers, namely, B3 (existing buildings were not designed for deconstruction), B7 (lack of information sharing, cooperation, and coordination among SC entities), B9 (lack of technical support, standard codes, and regulations in favor of using recycled materials) and B12 (lack of knowledge about RL in the construction industry). Finally, at the bottom of the ISM model, level IV, the highest hierarchical level, includes barrier B10 (lack of financial incentives to incorporate recycled

materials into the construction); this barrier influences all the other barriers and, as such, it is considered the key barrier to the ARLC in Portugal.

Lastly, at the bottom of the ISM model, level IV and level III, there are two barriers related to public policy, reflecting the relevance of this type of barriers to the ARLC [65]. Moreover, the position of the barriers C3, C7, and C12 at level III, the second-highest, reveals the importance of the operational-related barriers. On the contrary, the cost-related and organizational-related barriers are positioned at the top of the ISM model at level I [67]. These barriers are largely influenced by the other barriers and play a minor role in developing measures to mitigate the ARLC.

### 4.2. MIMAC Diagram

In phase III of the research methodology (Figure 1), the MICMAC diagram was built with the DPP and DVP of the barriers (Table 6) to evaluate further their impact on the ARLC. The resulting DPP-DVP diagram shows the barriers allocated to one of the clusters (Figure 3).

The Linkage and Autonomous clusters have no barrier allocated. The existence of MICMAC diagrams with empty clusters in the literature is nothing new. For example, in Ribeiro et al. [68], the Linkage cluster is empty, and in Senna et al. [69,70], the Autonomous cluster is empty. The barriers in the Autonomous cluster would present low DPP and DVP, meaning they would be somehow disconnected from the remaining barriers. Therefore, the inexistence of barriers in this cluster means that all barriers to the ARLC form a strongly interlinked system.

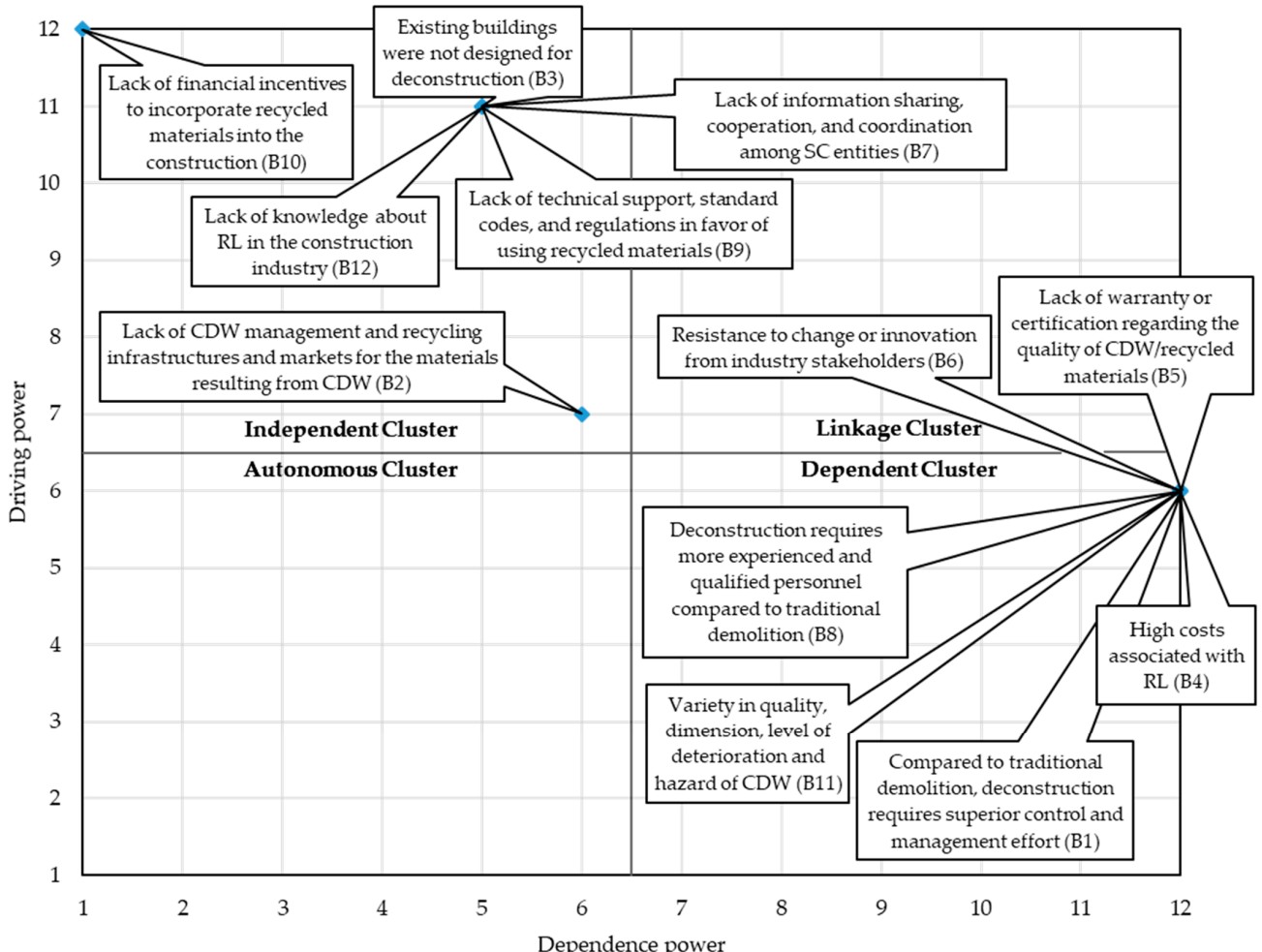

**Figure 3.** The MICMAC diagram of the barriers to the ARLC in Portugal.

The Dependent cluster comprises six barriers. Namely, deconstruction requires superior control and management effort compared to traditional demolition (B1), high costs associated with RL (B4), lack of warranty or certification regarding the quality of CDW/recycled materials (B5), resistance to change or innovation from industry stakeholders (B6), deconstruction requires more experienced and qualified personnel compared to traditional demolition (B8), and variety in the quality, dimension, level of deterioration, and hazard of CDW (B11). The barriers in this cluster are characterized by low DVP and high DPP. Therefore, they have a minor influence on the other barriers, and, in the present case, due to the absence of barriers in the Linkage cluster, they are directly influenced by the barriers of the independent cluster.

Finally, the Independent cluster contains six barriers. Namely, lack of CDW management and recycling infrastructures and markets for the materials resulting from CDW (B2), existing buildings were not designed for deconstruction (B3), lack of information sharing, cooperation, and coordination among SC entities (B7), lack of technical support, standard codes and regulations in favor of using recycled materials (B9), lack of financial incentives to incorporate recycled materials (B10), and lack of knowledge about RL in the construction (B12). These barriers present high DVP and low DPP; consequently, they influence the other barriers and are practically not influenced by them. Thus, they are considered the key barriers to the ARLC in Portugal.

## 5. Discussion and Mitigations Measures

In this section, with the help of the experts in GF3, the ISM model, and the MICMAC diagram, the mitigation measures are presented and discussed (phase IV of the research methodology). The combination of the hierarchical structure of interrelationships between barriers and their DVP and DPP allows for capitalizing on the discussion about the role of the barriers in the ARLC in developing mitigation measures. Therefore, the key barriers from the ISM model (at level IV) and the key barriers from the MICMAC diagram (in the Independent cluster) are labeled as root barriers to the ARLC in Portugal, namely, B2, B3, B7, B9, B10, and B12. These barriers influence the remaining barriers, thus representing the major constraints to the ARLC. However, although located in the same MICMAC cluster, it cannot be disregarded that these root barriers are in different hierarchical levels in the ISM model.

The experts were instructed to design measures, preventive in nature, to mitigate the root barriers, considering the hierarchy between them, and, if needed, additional measures to mitigate the other barriers in the Dependent cluster and at level I of the ISM model (B1, B4, B5, B6, B8, B11). The assumptions behind these instructions are: as the barriers are strongly interconnected, the preventive measures developed to act and mitigate the root barriers must also be designed to reach and mitigate the remaining hierarchically dependent barriers; if the previous measure failed to effectively mitigate the barriers at the dependent cluster, some additional measures should be prepared in advance.

The barrier of a lack of financial incentives to incorporate recycled materials into the construction (B10), at the highest hierarchal level of the ISM model, is the barrier with the most influence on the system. A governmental financial incentive for incorporating recycled materials to the detriment of virgin materials is probably the more practical and effective measure compared to the obligation imposed by legislation and regulation and, therefore, with a more significant impact on behavior change [3,60]. According to the results, the lack of financial incentives directly influences the barriers at level III. These incentives would: reduce the impact of the aggravated costs of the RL operations with the existing buildings that were not designed for deconstruction (B3); motivate and promote deeper collaboration among entities, thus increasing the information sharing, cooperation, and coordination among SC entities (B7); compel the government (together with the other entities) to provide technical support and develop standard codes, and regulations in favor of using recycled materials (B9); and, multiply the cases of successful adoption of RL in the construction industry, reducing the impact of the barrier lack of knowledge about RL

in the construction industry (B12). As mitigation measures, the experts recommended tax reductions for companies that incorporate recycled materials in their projects [38] and additional taxes on virgin materials to encourage using CDW-related materials, making them more competitive in the market [54]. However, this measure should be carefully studied because it can increase construction costs when using new materials without any environmental advantage. To avoid this undesirable effect, the experts also suggested that the government provide subsidies or tax incentives to companies that use and promote the deconstruction of existing buildings and guarantee CDW-related quality materials [71].

Concerning the barrier of existing buildings that are not designed for deconstruction (B3), although several authors refer to construction principles that should be included when designing for deconstruction, the experts agree that the construction industry is still unaware of those principles [9]. Guy et al. [72] suggest that the experience of deconstructing existing buildings should be used as an opportunity for learning, increasing knowledge, and developing the concept of design for deconstruction. Moreover, increasing landfill taxes, introducing restrictions on landfilling, and creating minimum percentages for CDW material recovery can make the deconstruction of buildings a more common practice [9,71]. More flexible legislation concerning CDW-related materials is also critical in promoting the deconstruction of existing buildings and designing new ones considering deconstruction. Kanters [73] mentions that making clients aware of the potential environmental and economic benefits could also encourage design for deconstruction.

The barrier of lack of information sharing, cooperation, and coordination among SC entities (B7) contributes to diminishing confidence in the quality and performance of CDW-related materials, consequently decreasing their demand and use in the construction project [39,54]. Requiring a deconstruction plan before issuing a license or permit can effectively improve cooperation and coordination between the entities involved. This measure would also mean that the relevant information is made available and accessible at a preliminary phase of the project to all involved entities. However, it is not enough to have the information available. It is also necessary to have information systems to manage and share this information [29,40]. Organizing and regulating markets for salvaged and recycled materials through electronic platforms requiring information sharing among entities could improve cooperation and coordination because they would have to align efforts to generate new economic opportunities [38]. Promoting workshops and other initiatives on RL could promote more contact between entities and future cooperation/coordination relations.

Due to the lack of technical support, standard codes, and regulations in favor of using recycled materials (B9), the use of these materials in construction projects is still occasional [3,38]. The lack of experience of designers and contractors in dealing with these CDW-related materials increases the uncertainty about their possible on-site applications and their respective performances. An essential aspect of increasing the use of these materials on-site is simplifying existing legislation and elaborating appropriate standards and technical specifications [32]. Thus, existing technical support, standard codes, and regulatory measures should be reviewed and simplified to facilitate the use of recycled materials and minimize doubts and errors in their interpretation. Additionally, additional standard codes and technical support should be created regarding other applications for these materials in the construction industry. Yet another suggestion from the experts was that legislation to incorporate CDW-related materials in private construction projects should also be implemented, similar to the existing one for public works. Although, it should be noted that economic-financial and information and awareness instruments should also be considered to strengthen this measure.

Concerning the lack of knowledge about RL in the construction industry (B12), the construction industry has shown some reluctance to innovate in terms of deconstruction, designing for deconstruction, and the incorporation of CDW-related materials while remaining very focused on the price competition strategies [74]. Moreover, the viability of RL depends on the type of materials used in new buildings since designers can reuse CDW-related materials, thus promoting the deconstruction of existing buildings, promoting

reuse and recycling rates, and advancing the resale markets [4]. Consequently, the lack of commitment to deconstruction and the inexistence of successful cases of RL in the industry have further hampered the development of RL and related knowledge, thus creating a vicious cycle. Thus, the experts referred to the importance of sharing information on good practices of RL and CDW management and development studies and technical support to validate/ensure the quality of CDW-related materials.

The barrier of a lack of CDW management and recycling infrastructures and markets for the materials resulting from CDW (B2) is located at level II of the ISM model and is directly influenced by B3, B7, B9, and B12 and, as such, is already partially targeted by some of the so far proposed mitigation measures. Nevertheless, the experts suggested that the number of management and recycling infrastructures should be increased and strategically located, taking into account anticipated deconstruction sites and the location of potential markets to minimize transport costs.

Lastly, the experts verified that the proposed mitigation measures (gathered in Table 9) reach and mitigate the barriers in the dependent cluster. Namely, deconstruction requires superior control and management effort compared to traditional demolition (B1), high costs associated with RL (B4), lack of warranty or certification regarding the quality of CDW/recycled materials (B5), resistance to change or innovation from industry stakeholders (B6), deconstruction requires more experienced and qualified personnel compared to traditional demolition (B8), and variety in the quality, dimension, level of deterioration, and hazard of CDW (B11).

**Table 9.** Mitigation measures for barriers to the ARLC in Portugal.

| Code | Mitigation Measures |
| --- | --- |
| M1 | Tax reductions for companies that incorporate recycled materials in their construction projects |
| M2 | Additional taxes on virgin materials to encourage using CDW-related materials |
| M3 | Reduction of taxes or subsidies for companies that use and promote the deconstruction of existing buildings and guarantee quality CDW-related materials |
| M4 | Requiring a deconstruction plan before issuing a license or permit |
| M5 | Information systems to manage and share information concerning RL operations |
| M6 | Organizing and regulating markets for salvaged and recycled materials through an electronic platform |
| M7 | Promote workshops and other initiatives on RL in the construction industry |
| M8 | Promote training courses on deconstruction based on practical examples of deconstructing existing buildings |
| M10 | Increase landfill taxes, introduce restrictions on landfilling, and create minimum percentages for material recovery |
| M11 | Making the legislation concerning the use of CDW-related materials more flexible |
| M12 | Launch client awareness campaigns for the potential environmental and economic benefits of using CDW-related materials in construction |
| M13 | Review and simplification of existing technical support, standard codes, and regulations regarding the use of CDW-related materials, and the creation of new ones |
| M14 | Creation of legislation to mandate the incorporation of CDW-related materials in private construction projects |
| M15 | Promote the sharing of information on good practices of RL and CDW management |
| M16 | Developing studies and technical support to validate/ensure the quality of CDW-related materials |
| M17 | Increase the number of management and recycling infrastructures strategically located considering the deconstruction sites and the markets |

## 6. Conclusions

Despite the economic, environmental, and social benefits related to RL, its level of implementation in the Portuguese construction industry is still low. This is not exclusive to this market but is a general pitfall of the construction industry worldwide. The root barriers to the implementation of RL were identified as follows: lack of financial incentives to incorporate recycled materials into the construction (B10), existing buildings that were not designed for deconstruction (B3), lack of information sharing, cooperation, and coordination among SC entities (B7), lack of technical support, standard codes, and regulations in favor of using recycled materials (B9), lack of knowledge about RL in the construction industry (B12), and lack of CDW management and recycling infrastructures and markets

for the materials resulting from CDW (B2). B10 and B9 are related to public policies, while B3, B7, B12, and B2 are related to operational aspects. This means that the success of the implementation of RL in the construction industry is highly dependent on governments' and companies' motivation to mitigate these barriers' impact. It is important to highlight that there are some certificates, e.g., Building Research Establishment's Environmental Assessment Method (BREEAM) and Building Research Establishment Global (BRE), which somehow provide an evaluation of the quality of the construction, including, among other dimensions, the management of "waste". However, these evaluation mechanisms have not been able, on their own, to provide the necessary incentives to investors and developers.

As the determination of critical barriers and the contextualization of the interrelationship among barriers was based on the experts' perception of importance, this may present some degree of subjectivity in the ISM-MICMAC approach. So, future studies should be carried out to confirm and reinforce the findings of the present studies, particularly in different markets. It is also recommended to consider applying qualitative methods to determine the degree of influence between critical barriers, as one barrier can influence many, but its influence can be low.

**Author Contributions:** Conceptualization, M.P., A.A., C.O.C.; Methodology and Literature Retrieval, M.P., A.A., C.O.C.; Writing, Original Draft Preparation, M.P., A.A.; Writing, Review, and Editing, A.A., C.O.C.; Supervision, A.A., C.O.C. All authors have read and agreed to the published version of the manuscript.

**Funding:** This research was fully funded by FCT—Fundação para a Ciencia e Tecnologia (Portugal), national funding through research grant UIDB/04625/2020.

**Informed Consent Statement:** Informed consent was obtained from all subjects involved in the study.

**Data Availability Statement:** Some or all data, models, or codes that support the findings of this study are available from the corresponding author upon reasonable request.

**Acknowledgments:** The authors would like to acknowledge all the experts involved in the focus group discussions.

**Conflicts of Interest:** The authors declare no conflict of interest.

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
