# Peer review of "Barriers to the Adoption of Reverse Logistics in the Construction Industry: A Combined ISM and MICMAC Approach"

_sustainability, doi:10.3390/su142315786_

Round 1

Reviewer 1 Report

The paper "Barriers to the Adoption of Reverse Logistics in the Construction Industry: A combined ISM and MICMAC Approach" falls within the scope of the journal "Sustainability"  but doesn't meet the standard quality of the paper that should be published in one prestigious journal in the current version. Some elements of one well-written and performed study are missing, so the paper needs improvements. 

1) The abstract is too long. Should be shorter with an emphasis on novelty, aims, contributions, and results.

2) The first section should be rewritten and extended:

- In the introduction section the following tasks should be fulfilled: the introduction should give an overview of the field's significance, and should consider the following main questions: What are the gaps in literature? What are the main aims of this article?" Also, contributions should be better described.

3) The second section Literature review is very good, but can be updated with the following sources new dates:

Tyagi, M., Panchal, D., Kumar, D., & Walia, R. S. (2021). Modeling and Analysis of Lean Manufacturing Strategies Using ISM-Fuzzy MICMAC Approach. Operational Research in Engineering Sciences: Theory and Applications, 4(1), 38-66.

- Wardito, E., Purba, H. H., & Purba, A. (2021). System dynamic modeling of risk management in construction projects: A systematic literature review. Operational research in engineering sciences: theory and applications, 4(1), 1-18.

4) Technical quality of the paper should be improved. Different font size...

5) Tables can be more explained.

6) Please explain in more detail the reasons for deciding to apply such a methodology. 

7) Figure 2 is low quality.

Author Response

The paper "Barriers to the Adoption of Reverse Logistics in the Construction Industry: A combined ISM and MICMAC Approach" falls within the scope of the journal "Sustainability"  but doesn't meet the standard quality of the paper that should be published in one prestigious journal in the current version. Some elements of one well-written and performed study are missing, so the paper needs improvements.

1) The abstract is too long. Should be shorter with an emphasis on novelty, aims, contributions, and results.

Reply:

Thank you for the comment. The abstract was pruned and now reads as follows:

ABSTRACT

With the growing environmental concerns, reverse logistics (RL) assumes a key role in the sustainability of the construction industry to the extent that it can contribute to mitigating some of the negative environmental impacts related to its activity. However, despite the benefits that can be attributed to RL, its implementation level in the construction industry is still very low. This research determines the root barriers to adopting reverse logistics in construction (ARLC) using the case of the Portuguese construction market. The methodology involved focus groups, Interpretive Structural Modelling (ISM) approach and Matrices d’Impacts cross-multiplication appliqúe a classmate (MICMAC). Based on the results, the root barriers to the implementation of RL in the Portuguese construction industry are: lack of financial incentives to incorporate recycled materials, lack of knowledge about RL, lack of technical support, standard codes and regulations in favor of using recycled materials, lack of information sharing, cooperation and coordination among entities of the supply chain, existing buildings have not been designed for deconstruction and lack of construction and demolition waste (CDW) management and recycling infrastructures and markets for the materials resulting from CDW.

2) The first section should be rewritten and extended:

- In the introduction section the following tasks should be fulfilled: the introduction should give an overview of the field's significance, and should consider the following main questions: What are the gaps in literature? What are the main aims of this article?" Also, contributions should be better described.

Reply:

We have rewritten part of the first section, highlighting the main research question, main findings, the methodology adopted, and the literature's novelty.

3) The second section Literature review is very good, but can be updated with the following sources new dates:

Tyagi, M., Panchal, D., Kumar, D., & Walia, R. S. (2021). Modeling and Analysis of Lean Manufacturing Strategies Using ISM-Fuzzy MICMAC Approach. Operational Research in Engineering Sciences: Theory and Applications, 4(1), 38-66.

- Wardito, E., Purba, H. H., & Purba, A. (2021). System dynamic modeling of risk management in construction projects: A systematic literature review. Operational research in engineering sciences: theory and applications, 4(1), 1-18.

Reply:

We have added both papers.

4) Technical quality of the paper should be improved. Different font size...

Reply:

We have verified and corrected the manuscript following the journal (Sustainability) template.

5) Tables can be more explained.

Reply:

We added some clarifications and explanations about the Tables to the manuscript.

6) Please explain in more detail the reasons for deciding to apply such a methodology.

Reply:

We have rewritten the first section and highlighted the choice of methodology.

7) Figure 2 is low quality.

Reply:

We have improved the quality of Figure 2. Moreover, we also improve the quality of Figure 3.

Reviewer 2 Report

The article has a well-sounding and unambiguous title in order for the reader to obtain a correct perception of the intentions and achieved research results contained in the article. The undertaken subject very well extends the issues of Reverce Logistics to include the service sector of the construction industry. Abstract, although descriptive, gives a good overview of the research methodology used and the results achieved. The differences in the terminology of “reuse or recycling” [14] were rightly emphasized, but the construction industry is presented on the basis of the “construction (new objects) by destruction (old objects) approach. However, after an in-depth reading, it can be seen that it fits in with the general idea of "Sustainable Development" through the recovery of materials as well as reclamation and re-development of a specific area, especially in urban space. I also appreciate the adopted good methodological and formal structure and the way of organizing the entire text, in the form of: IMRaD - Introduction, Methods, Results and Discussion. In Keywords, I would consider the term "mitigation measures", because in the context of the title, it is more about "overcome barriers" (corrective action), or then "elimination of barriers" (preventive measures), and it would probably be appropriate to fully expand the abbreviation "ISM” to “Interpretive Structural Modeling ”(although the term is expanded in Abstract). They analyze individual parts of the formal structure of the article (its structure), Introduction [33-67] is correct, as are the considerations contained in Bacground [70-340] about the nature of Reverse Logistics, key chalanges and barriers to adoption, especially through the use of appropriate and representative bibliographic footnotes and conducted a synthetic critical analysis of the literature on the subject. Part 3 proves that the properties of the adopted scientific approach are particularly correct, entitled Methods and materials [315-466], especially the illustrative Figure 1. Research methodology. Such a clear visualization of the research process with a justified division into individual phases (and steps) confirms the high methodological maturity in conducting scientific research represented by the authors. The conducted scientific arguments, systematics and visualization of the obtained research results contained in Results [467-533] confirm the high efficiency in using the developed research apparatus. The Discussion and mitigations measures area [554-664] is an objective reflection of the results obtained. However, already in Conlussion [665-685] some conclusions were written too categorically (e.g. "/ ... / Lack of financial incentives to incorporate recycled materials into the construction (B10) / ... /" [669-670], because there is e.g. Certificate BREEAM (Building Research Establishment's Environmental Assessment Method) Building Research Establishment Global (BRE), which is a system for assessing the quality of buildings, currently standard in Europe and in the world, whose scoring also includes the "Waste" category. In the final assessment, the article has a very high level of methodological maturity and well-established formal professionalism, therefore I recommend it for publication in "Sustainable".

Author Response

The article has a well-sounding and unambiguous title in order for the reader to obtain a correct perception of the intentions and achieved research results contained in the article. The undertaken subject very well extends the issues of Reverce Logistics to include the service sector of the construction industry. Abstract, although descriptive, gives a good overview of the research methodology used and the results achieved. The differences in the terminology of “reuse or recycling” [14] were rightly emphasized, but the construction industry is presented on the basis of the “construction (new objects) by destruction (old objects) approach. However, after an in-depth reading, it can be seen that it fits in with the general idea of "Sustainable Development" through the recovery of materials as well as reclamation and re-development of a specific area, especially in urban space. I also appreciate the adopted good methodological and formal structure and the way of organizing the entire text, in the form of: IMRaD - Introduction, Methods, Results and Discussion.

Reply:

Thank you for the overall analysis of our paper.

In Keywords, I would consider the term "mitigation measures", because in the context of the title, it is more about "overcome barriers" (corrective action), or then "elimination of barriers" (preventive measures), and it would probably be appropriate to fully expand the abbreviation "ISM” to “Interpretive Structural Modeling ”(although the term is expanded in Abstract).

Reply:

We have added the keyword suggested by the reviewer. We were not able to expand the abbreviations of the title, given the maximum length allowed.

They analyze individual parts of the formal structure of the article (its structure), Introduction [33-67] is correct, as are the considerations contained in Bacground [70-340] about the nature of Reverse Logistics, key chalanges and barriers to adoption, especially through the use of appropriate and representative bibliographic footnotes and conducted a synthetic critical analysis of the literature on the subject. Part 3 proves that the properties of the adopted scientific approach are particularly correct, entitled Methods and materials [315-466], especially the illustrative Figure 1. Research methodology. Such a clear visualization of the research process with a justified division into individual phases (and steps) confirms the high methodological maturity in conducting scientific research represented by the authors. The conducted scientific arguments, systematics and visualization of the obtained research results contained in Results [467-533] confirm the high efficiency in using the developed research apparatus. The Discussion and mitigations measures area [554-664] is an objective reflection of the results obtained. However, already in Conlussion [665-685] some conclusions were written too categorically (e.g. "/ ... / Lack of financial incentives to incorporate recycled materials into the construction (B10) / ... /" [669-670], because there is e.g. Certificate BREEAM (Building Research Establishment's Environmental Assessment Method) Building Research Establishment Global (BRE), which is a system for assessing the quality of buildings, currently standard in Europe and in the world, whose scoring also includes the "Waste" category.

Reply:

We have expanded the discussion and added the example given by the reviewer.

In the final assessment, the article has a very high level of methodological maturity and well-established formal professionalism, therefore I recommend it for publication in "Sustainable".

Reply:

Thank you for the evaluation.

Reviewer 3 Report

The problem has not been well-descripted in the first section, Introduction.

Literature review is very long. Cut reviewed articles in half.
Please check and cite following related paper:

Asghari, M.; Abrishami, S.J.; Mahdavi, F., Reverse logistics network design with incentive-dependent return. Industrial Engineering and Management Systems. 2014, 13 (4), 383-397, doi:10.7232/iems.2014.13.4.383

·         Tables, figures and equations should be rearranged to fit the width of the pages.

The contribution of the research must be clearly explained compared to the existing works. 

Author Response

The problem has not been well-descripted in the first section, Introduction.

Reply:

We have improved the organization and description in the first section to clarify the problem (research questions) and methodology adopted

Literature review is very long. Cut reviewed articles in half.

Please check and cite following related paper:

Asghari, M.; Abrishami, S.J.; Mahdavi, F., Reverse logistics network design with incentive-dependent return. Industrial Engineering and Management Systems. 2014, 13 (4), 383-397, doi:10.7232/iems.2014.13.4.383

Reply:

We have added the paper, which helped justify/support a mitigation measure.

Tables, figures and equations should be rearranged to fit the width of the pages.

Reply:

Following the paper template, we have rearranged the tables and figures to fit the width of the column (narrower) or page (wider). Concerning equation (1), we already follow the paper template.

The contribution of the research must be clearly explained compared to the existing works.

Reply:

We have rewritten the first section and highlighted the main contribution and novelty of the paper.

Round 2

Reviewer 1 Report

The paper has been revised according to my suggestions.

Author Response

Dear Review,

Thank you very much for your suggestions/ recommendations.

Your contribution was significant to the improvement of the manuscript.

Our best regards,

The authors

Reviewer 3 Report

Thanks to the authors for the revisions. The paper may be accepted.

Author Response

(The authors gave the same response as above.)
